# Management of Synchronous Extrathoracic Oligometastatic Non-Small Cell Lung Cancer

**DOI:** 10.3390/cancers13081893

**Published:** 2021-04-15

**Authors:** Gregory D. Jones, Harry B. Lengel, Meier Hsu, Kay See Tan, Raul Caso, Amanda Ghanie, James G. Connolly, Manjit S. Bains, Valerie W. Rusch, James Huang, Bernard J. Park, Daniel R. Gomez, David R. Jones, Gaetano Rocco

**Affiliations:** 1Thoracic Service, Department of Surgery, Memorial Sloan Kettering Cancer Center, New York, NY 10065, USA; gdj9003@nyp.org (G.D.J.); lengelh@mskcc.org (H.B.L.); Raul.Caso@gunet.georgetown.edu (R.C.); connolj1@mskcc.org (J.G.C.); bainsm@mskcc.org (M.S.B.); ruschv@mskcc.org (V.W.R.); huangj@mskcc.org (J.H.); parkb@mskcc.org (B.J.P.); jonesd2@mskcc.org (D.R.J.); 2Department of Epidemiology and Biostatistics, Memorial Sloan Kettering Cancer Center, New York, NY 10065, USA; hsum1@mskcc.org (M.H.); tank@mskcc.org (K.S.T.); 3College of Medicine, SUNY Upstate Medical University, Syracuse, NY 13210, USA; Amanda.ghanie@gmail.com; 4Druckenmiller Center for Lung Cancer Research, Memorial Sloan Kettering Cancer Center, New York, NY 10065, USA; gomezd@mskcc.org; 5Department of Radiation Oncology, Memorial Sloan Kettering Cancer Center, New York, NY 10065, USA

**Keywords:** oligometastasis, non-small cell lung cancer, primary tumor resection

## Abstract

**Simple Summary:**

Although oligometastatic disease is common, present in up to 25% of patients with stage IV non-small cell lung cancer, management of it remains challenging. Numerous other studies have shown promising results in patients who undergo local treatment of both the primary tumor and the metastases. In this, the largest single-institution analysis of patients undergoing primary tumor surgical resection for oligometastatic disease, we have demonstrated encouraging long-term event-free survival, overall survival, and postrecurrence survival, with the greatest benefit in patients who undergo neoadjuvant therapy and those with limited intrathoracic disease. Therefore, in carefully selected patients, surgical resection of the primary tumor can be an important component of multimodal management for advanced-stage non-small cell lung cancer.

**Abstract:**

Stage IV non-small cell lung cancer (NSCLC) accounts for 35 to 40% of newly diagnosed cases of NSCLC. The oligometastatic state—≤5 extrathoracic metastatic lesions in ≤3 organs—is present in ~25% of patients with stage IV disease and is associated with markedly improved outcomes. We retrospectively identified patients with extrathoracic oligometastatic NSCLC who underwent primary tumor resection at our institution from 2000 to 2018. Event-free survival (EFS) and overall survival (OS) were estimated using the Kaplan–Meier method. Factors associated with EFS and OS were determined using Cox regression. In total, 111 patients with oligometastatic NSCLC underwent primary tumor resection; 87 (78%) had a single metastatic lesion. Local consolidative therapy for metastases was performed in 93 patients (84%). Seventy-seven patients experienced recurrence or progression. The five-year EFS was 19% (95% confidence interval (CI), 12–29%), and the five-year OS was 36% (95% CI, 27–50%). Factors independently associated with EFS were primary tumor size (hazard ratio (HR), 1.15 (95% CI, 1.03–1.29); *p =* 0.014) and lymphovascular invasion (HR, 1.73 (95% CI, 1.06–2.84); *p =* 0.029). Factors independently associated with OS were neoadjuvant therapy (HR, 0.43 (95% CI, 0.24–0.77); *p =* 0.004), primary tumor size (HR, 1.18 (95% CI, 1.02–1.35); *p =* 0.023), pathologic nodal disease (HR, 1.83 (95% CI, 1.05–3.20); *p =* 0.033), and visceral-pleural invasion (HR, 1.93 (95% CI, 1.10–3.40); *p =* 0.022). Primary tumor resection represents an important treatment option in the multimodal management of extrathoracic oligometastatic NSCLC. Encouraging long-term survival can be achieved in carefully selected patients, including those who received neoadjuvant therapy and those with limited intrathoracic disease.

## 1. Introduction

Stage IV non-small cell lung cancer (NSCLC) accounts for 35 to 40% of all newly diagnosed cases of NSCLC [1,2]. Despite the evolution of precision medicine and targeted systemic therapies, the prognosis remains dismal for patients with metastatic NSCLC, with an estimated two-year survival rate of 10 to 23% and a five-year survival of 0% to 10% [3]. The oligometastatic state—defined as ≤5 extrathoracic metastatic lesions in ≤3 organs [4]—is present in approximately 25% of patients with stage IV disease [5] and, compared with more-extensive disease, is associated with markedly improved outcomes, with a five-year survival up to 30% [6].

Although aggressive therapy aimed at eliminating all metastatic sites has been shown to lead to durable disease control in other cancers (e.g., colorectal carcinoma, melanoma, and sarcoma) [7], this treatment paradigm remains controversial in metastatic NSCLC, for which definitive systemic therapy with or without radiotherapy continues to be the cornerstone of management. However, a landmark phase II clinical trial of 49 patients with oligometastatic NSCLC demonstrated an eight-month increase in progression-free survival (PFS) in patients who underwent local consolidative therapy (LCT), defined as treatment with the goal to ablate or resect all residual disease using radiotherapy or surgery, for all disease sites, compared with patients who underwent only maintenance or observation (median PFS, 11.9 vs. 3.9 months; *p* = 0.005) [8]. The subsequent follow-up to this multicenter, randomized trial affirmed this benefit in PFS and observed an additional overall survival (OS) benefit of 24.2 months (*p =* 0.017) [9].

Although more-recent prospective, randomized trials have since confirmed the survival benefit afforded by LCT for all disease sites [10,11], none has included patients who underwent surgical resection of the primary tumor. Additionally, although the use of chemotherapy and targeted systemic therapies continues to increase, a recent analysis of >23,000 patients with oligometastatic NSCLC revealed that the utilization of primary tumor resection significantly decreased over the eight-year study period [12]. For these reasons, the role of surgery in this context requires further exploration. In this study, we investigated event-free survival (EFS) and OS, and determined factors associated with these outcomes, in our institutional cohort of patients who underwent primary tumor resection in the management of oligometastatic NSCLC.

## 2. Materials and Methods

### 2.1. Patients

After approval from our institutional review board, with the need for patient consent waived, we performed a retrospective review of our prospectively maintained database to identify all patients with clinical stage IV NSCLC at the time of diagnosis who underwent resection of the primary tumor with and without LCT for synchronous metastases between 2000 and 2018. The oligometastatic state was defined as up to 5 extrathoracic lesions in up to 3 organs at the time of diagnosis, in accordance with the published consensus definition [4]. Patients who had synchronous primary tumors, who underwent primary tumor surgery without curative intent (i.e., for palliative or diagnostic purposes), or who had >5 extrathoracic lesions at diagnosis were excluded (Appendix A). Synchronous primary tumors were distinguished from pulmonary metastases using Martini and Melamed criteria, with confirmation of clonal relatedness using genomic data whenever possible, as previously described by our group [13]. Patients who did not undergo LCT for metastatic lesions were given standard of care systemic therapy in the neoadjuvant or adjuvant setting with resection of the primary tumor and were included in the survival analyses. Patients underwent LCT for metastatic lesions either before or after resection of the primary tumor, with restaging at the time of primary tumor resection. Oligometastatic sites of disease were broadly categorized as adrenal gland, liver, bone, brain, multiple, and other. The other classification included the following sites: eye, intestine, omentum, pancreas, scalp, spleen, and perianal soft tissue.

### 2.2. Management of Patients with Oligometastatic Disease

All patients were clinically staged with computed tomography and positron emission tomography scans. Although the National Comprehensive Cancer Network guidelines [14] do not require invasive mediastinal staging for patients with metastatic NSCLC, most patients underwent mediastinal staging via endobronchial ultrasound; the remainder were clinically node-negative on preoperative imaging. Systemic therapy (preoperative, postoperative, or both) was administered using standard of care first-line regimens, with the choice of specific agents at the discretion of the treating medical oncologist. The time from LCT for metastases to primary tumor resection was calculated from the date of completion of LCT for metastases. For patients considered for primary lung resection, the radiologic response to neoadjuvant therapy was characterized by Response Evaluation Criteria in Solid Tumours (RECIST) [15] for both the primary tumor and the extrathoracic sites of disease. Surgical resection included segmentectomy or lobectomy, with mediastinal lymph node sampling. The follow-up was conducted in accordance with the National Comprehensive Cancer Network guidelines [14], and the timing and location of all recurrences were recorded. The progression of disease was radiologically defined in accordance with RECIST 1.1 guidelines [15]. The sites of recurrence or progression were again classified into adrenal gland, liver, bone, brain, multiple, and other. The other classification included abdominopelvic lymph nodes, kidney, and uterus.

### 2.3. Outcomes and Statistical Analysis

The primary outcomes in this study were EFS and OS. EFS was defined as the time to recurrence, progression, or death and was calculated from the date of primary lung tumor resection. OS was defined as the time from primary lung tumor resection to death. Patients were otherwise censored at the time of last follow-up. EFS and OS were estimated using the Kaplan–Meier method. Overall survival after site-specific recurrence or progression was also estimated using the Kaplan–Meier method and was compared between the sites listed above using the log-rank test. The median follow-up duration was calculated using the reverse Kaplan–Meier method [16]. The factors associated with EFS and OS were determined using Cox proportional hazards regression with hazard ratios (HRs) and 95% confidence intervals (CIs). Multivariable models were constructed using backwards elimination, starting with all factors with *p* < 0.1 on univariable analysis. Statistical tests were 2-sided, and *p* < 0.05 was considered statistically significant. Statistical analyses were conducted using R 3.5.2 (R Development Core Team, Vienna, Austria).

## 3. Results

### 3.1. Patients, Tumor Characteristics, and Treatment Details

A total of 111 patients with oligometastatic NSCLC (total metastatic lesions, 147) underwent primary tumor resection during the study period (Table 1). The median age at surgery was 62 years (interquartile range (IQR), 53–69 years), and the majority of patients (*n =* 65 (59%)) were women.

Of the 111 patients included, 101 (91%) had a single metastatic site, and 87 (78%) had a single metastatic lesion. The most common site of metastasis was the brain (*n =* 57 (51%)), followed by bone (*n =* 21 (19%)), liver (*n =* 8 (7%)), and adrenal gland (*n =* 6 (5%)). LCT for all metastatic lesions was performed in 93 patients (84%; total lesions, 106); treatment modalities included surgery alone (*n =* 28 (25%)), radiotherapy alone (*n =* 40 (36%)), and combined surgery and radiotherapy (*n =* 25 (23%)). Of the patients who underwent LCT for metastases, 76 of 93 (82%) had undergone treatment of metastatic lesions before resection of the primary tumor, with a median time from LCT for metastases to primary tumor resection of 5.0 months (IQR, 1.6–10 months). Brain metastases were the most commonly treated lesions (57/63 (90%)), with radiotherapy used in all but seven cases (50/57 (88%)). Bone metastases were the second most commonly treated lesions (23/30 (77%)), with radiotherapy again used in the majority of cases (15/23 (65%)).

Most patients (99/111 (89%)) underwent systemic therapy, including chemotherapy, immunotherapy, and targeted therapy. Of these, most received neoadjuvant therapy only (55/99 (56%)); 11% received adjuvant therapy only (11/99), and 33% received both (33/99). Most primary tumor resections (*n =* 70 (63%)) were performed via a thoracotomy approach; lobectomy was performed in 85 patients (76%). The median pathologic primary tumor size was 2.5 cm (IQR, 1.6–3.6 cm).

### 3.2. Follow-Up and Survival

Individual disease courses and interventions by initial site of metastasis are depicted in Figure 1.

The median follow-up for the entire cohort was 4.54 years (IQR, 3.73–7.25 years). Seventy-seven patients had recurrence or progression, and 60 patients died during the study period. Twenty-three patients were without evidence of disease at last follow-up. The longest EFS and OS were intervals of 11.3 years and 11.7 years, respectively (Figure 1A, Patient 57). The longest ongoing survival without any recurrence or progression to date was 9.2 years (Figure 1D, patient 6). Overall, two- and five-year EFS were 31% (95% CI, 23–42%) and 19% (95% CI, 12–29%) (Figure 2A), respectively, and two- and five-year OS were 77% (95% CI, 69–86%) and 36% (95% CI, 27–50%) (Figure 2B), respectively. No differences were observed when comparing single metastatic sites—although patients with multiple sites at the time of diagnosis had some indication of lower-risk EFS and OS. However, the small sample sizes were too limited to make definitive conclusions on the effect of specific metastatic sites (Appendix A) or the type of LCT for metastasis (Appendix A) on EFS and OS.

### 3.3. Patterns of Treatment Failure

Among patients who experienced recurrence or progression within the observation period (*n =* 77), 16 (21%) experienced locoregional recurrence within the ipsilateral hemithorax; the remaining 61 (79%) experienced recurrence or progression at distant sites.

Of these patients, 48 (62%) had a recurrence at a new site, and 29 (38%) experienced progression at the site of the previously treated metastasis. Rates of locoregional recurrence were higher after sublobar resection than after lobectomy (25% vs. 11%), whereas rates of distant recurrence or progression were higher after lobectomy (60% vs. 39%). The most common distant site was the brain (28/77 (36%)), followed by bone (13/77 (17%)), and adrenal gland (7/77 (9%)). Six patients (8%) experienced recurrence or progression at multiple sites. The three-year post-recurrence overall survival for all 77 patients with recurrence or progression was 41% (95% CI, 30–56%), with a median post-recurrence OS of 2.18 years (95% CI, 1.61–3.11 years). Post-recurrence survival did not significantly differ by site of recurrence or progression (*p =* 0.57; Appendix A). However, patients with pN1 or pN2 disease had a higher proportion of both bone recurrence and brain recurrence, compared with patients with pN0 disease (23% vs. 13% (*p =* 0.4) and 47% vs. 39% (*p =* 0.6), respectively). Conversely, patients with pN1 or pN2 disease had a significantly lower proportion of intrathoracic than nonintrathoracic recurrence compared with patients with pN0 disease (13% vs. 36% (*p =* 0.035)). Importantly, however, given the relatively few events, this represents an exploratory analysis of the prognostic value of intrathoracic recurrence, and any conclusions should be interpreted with caution.

### 3.4. Factors Associated with EFS and OS

On univariable analysis, multiple sites of metastasis at diagnosis, the total number of metastatic lesions, the micropapillary or solid subtype, lymphovascular invasion, visceral pleural invasion, pathologic primary tumor size, and pathologic N stage were associated with EFS (*p* < 0.1; Appendix A). On multivariable analysis, increasing pathologic primary tumor size (HR, 1.15 (95% CI, 1.03–1.29); *p =* 0.014) and lymphovascular invasion (HR, 1.73 (95% CI, 1.06–2.84); *p =* 0.029) were independently associated with EFS (Table 2).

On univariable analysis, receipt of neoadjuvant therapy, multiple sites of metastasis at diagnosis, lymphovascular invasion, visceral pleural invasion, pathologic primary tumor size, and pathologic N stage were associated with OS (*p* < 0.1; Appendix A). On multivariable analysis, receipt of neoadjuvant therapy (HR, 0.43 (95% CI, 0.24–0.77); *p =* 0.004), increasing pathologic primary tumor size (HR, 1.18 (95% CI, 1.02–1.35); *p =* 0.023), pathologic nodal disease (HR, 1.83 (95% CI, 1.05–3.20); *p =* 0.033), and visceral pleural invasion (HR, 1.93 [95% CI, 1.10–3.40]; *p =* 0.022) were independently associated with OS (Table 2). Importantly, receipt of neoadjuvant therapy was associated with a median OS of 4.19 years (95% CI, 3.97–8.44) versus 2.84 years (95% CI, 1.80–5.63) in those who did not undergo neoadjuvant therapy. Patients who received neoadjuvant therapy had a five-year OS of 40% (95% CI, 29–57%), compared with 21% (95% CI, 9–50%) in patients who did not receive neoadjuvant therapy (*p =* 0.018; Figure 3).

## 4. Discussion

In the largest single-institution analysis to date of patients who underwent primary tumor resection as part of multimodal management for oligometastatic NSCLC, we reported five-year EFS and OS that surpass rates historically associated with metastatic disease [3]. We also showed that EFS intervals exceeding 10 years can be attained in select cases, with a majority of patients alive at two years post-recurrence. Factors representative of aggressive primary tumor biology (e.g., primary tumor size, N stage, and lymphovascular or visceral pleural invasion) were independently associated with both EFS and OS. Conversely, survival does not seem to differ by site of metastasis at diagnosis nor by LCT treatment modality for metastasis. Finally, patients who undergo neoadjuvant therapy may experience the best OS, with five-year OS twice as long in patients who underwent neoadjuvant therapy.

Up to one-quarter of patients with stage IV NSCLC present with oligometastasis [5], and numerous studies have reported encouraging survival estimates in patients who have undergone LCT for both the primary tumor and the metastases. The landmark phase II clinical trial by Gomez and colleagues (*n =* 49) [8] and the subsequent follow-up study [9] reported an increase in median PFS of 8.0 months and 9.8 months, respectively, in patients who underwent LCT for all disease sites, compared with maintenance or observation alone. However, this trial was limited by the overall low number of patients who received LCT (*n =* 25). Although no subsequent prospective studies have included surgical patients [10,11], larger, more-recent retrospective studies have reiterated the survival benefit in patients whose oligometastasis management included primary tumor resection [17,18,19]. Single-institution studies have observed five-year OS in the range of 32 to 48% in patients who underwent primary tumor resection and LCT for all disease sites, compared with 24% in patients who underwent radiotherapy for local disease control [18,19]. A multi-institutional collaboration from Switzerland that included 124 patients with oligometastatic NSCLC who underwent primary tumor resection observed a median PFS of 11 months, with an encouraging five-year OS of 36% [17]. Despite these findings, surgical management of the primary tumor is utilized in a fraction of cases of oligometastatic NSCLC [5,20,21], and rates of primary tumor resection in this population have actually decreased during the past decade [12,22]. A retrospective American Cancer Registry analysis of more than 23,000 patients with metastatic NSCLC revealed that the rate of primary tumor resection significantly decreased from 2004 to 2012, with a disproportionate increase in the use of systemic therapy [12]. It is likely that improvements in systemic therapies, precision medicine, and radiologic staging, as well as changes in referral patterns, have shaped initial treatment approaches. However, our study’s encouraging five-year EFS of 19% and five-year OS of 36% suggest that primary tumor resection is not only a viable option but is an important contributor to improved outcomes in the multidisciplinary management of patients with oligometastatic NSCLC.

Even in the presence of distant metastasis, tumor size and nodal disease remain primary determinants of both EFS and OS in patients with oligometastatic NSCLC. In retrospective studies of patients with single-organ [23] or multi-organ [19] oligometastatic NSCLC who underwent primary tumor resection, a pathologic T stage was shown to be a prognosticator of survival. In our study, pathologic tumor size was independently associated with both EFS and OS. Nodal disease was also indicative of poor EFS and OS in our study, as has been observed in numerous other studies [5,17,24,25]. However, in our cohort, specific patterns of recurrence were associated with pathologic nodal status, with pN0 patients having seemingly higher proportions of intrathoracic recurrence than pN1 or pN2 patients on exploratory analysis. The substantial influence of nodal status on outcomes in patients with oligometastatic NSCLC, with worse OS on univariable and multivariable analysis, highlights the importance of adequate mediastinal staging both pre- and intraoperatively in this cohort and suggests that patients with limited intrathoracic disease may derive the most benefit from surgical management of their disease.

Intuitively, an increasing number of metastatic lesions at diagnosis—a surrogate for disease burden—should be associated with worse outcomes. However, the relationship between the number of metastatic lesions and outcomes remains a matter of controversy. The largest meta-analysis of patients with oligometastatic NSCLC found that an increasing number of metastatic lesions was associated with worse PFS on univariable but not multivariable analysis [25], and numerous retrospective studies have reported no association between increasing number of metastatic lesions and survival [6,26]. In our study, an increasing number of metastatic lesions was actually associated with better OS on univariable analysis but was not associated with EFS or OS on multivariable analysis. Possible explanations for this finding are that a minority of the patients in our cohort (*n =* 24 (22%)) had >1 metastatic lesion at diagnosis, and these patients required more-extensive therapy. For instance, all but one patient with >1 metastatic lesion received neoadjuvant systemic therapy (23/24 (96%)), compared with only 75% of patients with one metastatic lesion (65/87). This suggests that the extent of treatment received may contribute more to outcomes than the number of extrathoracic lesions at diagnosis.

Receipt of neoadjuvant therapy, which was administered to 79% of patients in our cohort, was an independent prognosticator of improved OS, with a median OS of 4.19 years, compared with 2.84 years in those who did not undergo neoadjuvant therapy. The results of a database study of 2065 patients with metastatic NSCLC echoed our findings, with receipt of systemic therapy leading to substantially improved OS (HR, 0.3 (95% CI, 0.26–0.37); *p* < 0.001) [27]. Neoadjuvant therapy, which has the goal of eliminating micrometastatic and nodal disease, has consistently been shown to influence survival in patients with oligometastatic NSCLC. Thus, multimodal management that includes neoadjuvant therapy and surgical resection of the primary tumor may achieve the best long-term outcomes in this population.

Our study has several limitations. Although our robust and detailed collection of clinicopathologic, recurrence, and treatment data allowed us to thoroughly explore the outcomes of patients in our cohort, the types and extents of treatments administered varied between patients, possibly introducing unmeasured confounding in the assessment of long-term outcomes. Furthermore, although our cohort represented the largest single-institution analysis of patients who underwent resection of their primary tumor, it was a select group of patients with appropriate performance status and comorbidities, the majority of whom underwent neoadjuvant therapy, presumably without significant progression of disease. Therefore, it is unknown what proportion of patients with oligometastatic disease did not undergo surgery or how that number has changed over time. In addition, as most patients underwent neoadjuvant therapy, as opposed to adjuvant therapy, we were unable to analyze the effect of systemic treatment following resection on survival. Although all patients who received surgery after 2014 in our study underwent primary tumor sequencing, this group accounted for a minority of our cohort, precluding assessment of the influence of tumor genomics on outcomes. Additionally, as mentioned previously, although molecular studies were performed following 2014 with select patients receiving matched targeted therapy, the small number of patients receiving adjuvant therapy precluded any analysis of this group. Finally, our study lacked a reference group of patients who did not undergo primary tumor resection for comparison of long-term outcomes.

## 5. Conclusions

Although oligometastatic NSCLC remains a diverse and challenging entity, we have shown that patients who receive neoadjuvant therapy and those with limited intrathoracic disease, defined by nodal status, are most likely to benefit from a surgery-focused approach to their primary site of disease (Video S1). Although the majority of patients in our study were limited to one site of metastasis, an increasing number of metastatic lesions was not associated with worse outcomes. Meanwhile, patients with nodal disease experienced worse EFS and OS, highlighting the need for accurate mediastinal staging in this oligometastatic cohort. Therefore, in carefully selected patients, on the basis of the above criteria and operative risk (Appendix A), surgical resection of the primary tumor can achieve encouraging long-term EFS, OS, and post-recurrence survival and can be a critical component of multimodal management for advanced stage NSCLC.

## Figures and Tables

**Figure 1 cancers-13-01893-f001:**
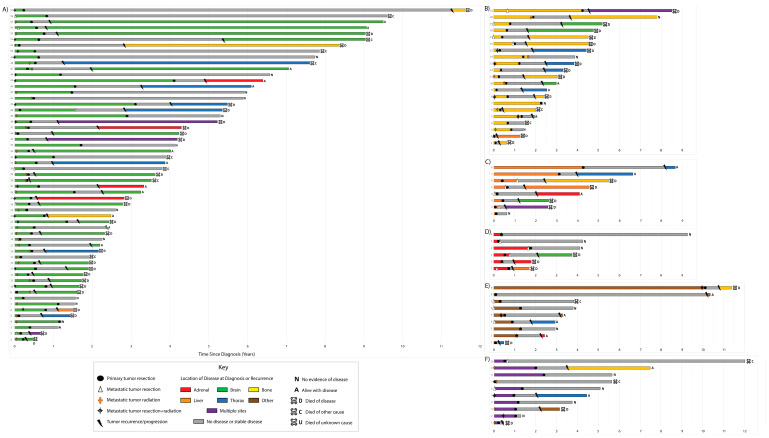
Swimmer plot depicting the disease course with recurrence and disease status and interventions, including primary resection and metastatic treatment, in our cohort, organized by initial site of metastasis: (**A**) brain, (**B**) bone, (**C**) liver, (**D**) adrenal gland, (**E**) other, (**F**) and multiple sites.

**Figure 2 cancers-13-01893-f002:**
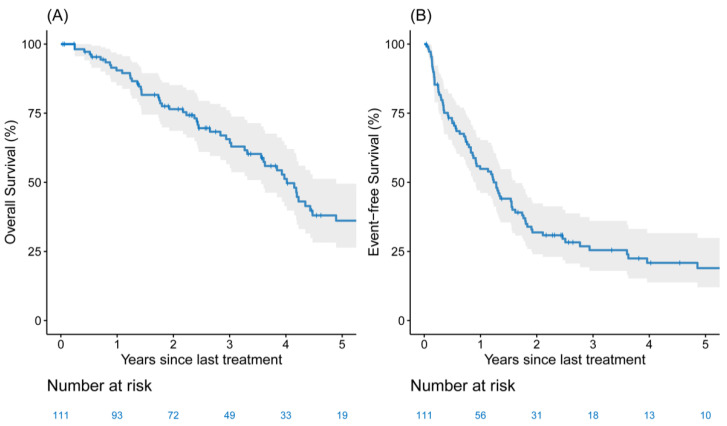
The Kaplan–Meier five-year (**A**) overall survival and (**B**) event-free survival estimates among patients (*n =* 111) with extrathoracic oligometastatic synchronous non-small cell lung cancer who underwent surgical resection of the primary tumor (solid lines are estimates; the shaded region forms the 95% confidence band).

**Figure 3 cancers-13-01893-f003:**
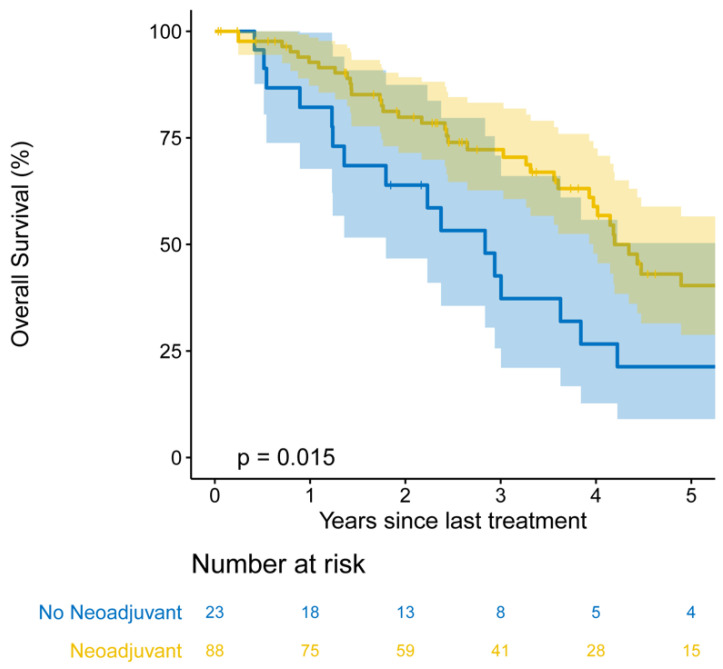
Kaplan–Meier five-year overall survival estimates among patients who received neoadjuvant therapy (*n =* 88) versus patients who did not (*n =* 23).

**Table 1 cancers-13-01893-t001:** Patient demographic, tumor, and treatment details (*n =* 111).

Variable	No. (%) or Median (IQR)
Age at surgery, years	62 (53–69)
Sex	
Female	65 (59)
Male	46 (41)
Smoking status	
Never	21 (19)
Ever	90 (81)
Pack-years	30 (4.5–48)
Radiologic primary tumor size, cm (*n =* 102)	3.3 (2.0–4.6)
Primary tumor SUVmax (*n =* 87)	9.6 (6.3–15)
FEV1, % (*n =* 103)	91 (79–101)
DLCO, % (*n =* 103)	82 (67–94)
Extrathoracic metastatic site at diagnosis	
Adrenal	6 (5)
Bone	21 (19)
Brain	57 (51)
Liver	8 (7)
Other ^a^	9 (8)
Multiple	10 (9)
Total metastatic sites	
1	101 (91)
2	8 (7)
3	2 (2)
Total metastatic lesions	
1	87 (78)
2	15 (14)
3	7 (6)
4	1 (1)
5	1 (1)
Local consolidative therapy for metastasis	
No	18 (16)
Yes	93 (84)
Neoadjuvant therapy	
None	23 (21)
Systemic therapy only ^b^	77 (69)
Chemoradiotherapy	11 (10)
Operative approach to primary tumor	
Open	70 (63)
VATS	41 (37)
Primary tumor resection type	
Lobectomy	85 (77)
Segmentectomy	26 (23)
Histologic subtype	
Lepidic	0 (0)
Acinar/papillary	30 (27)
Micropapillary/solid	14 (13)
Unknown	67 (60)
Final pathologic diagnosis	
Adenocarcinoma	80 (72)
Squamous cell carcinoma	6 (5)
Other	14 (13)
No viable tumor	11 (10)
Lymphovascular invasion	
No	45 (41)
Yes	60 (54)
Unknown	6 (5)
Visceral pleural invasion	
No	62 (56)
Yes	45 (41)
Unknown	4 (4)
Pathologic primary tumor size, cm	2.5 (1.6–3.6)
Pathologic stage (AJCC 8th edition) ^c^	
I	5 (5)
II	2 (2)
III	2 (2)
IV	102 (92)
Adjuvant therapy	
None	67 (60)
Systemic therapy only ^b^	31 (28)
Radiotherapy only	8 (7)
Chemoradiotherapy	5 (5)

AJCC, American Joint Committee on Cancer; DLCO, diffusion capacity of the lungs for carbon monoxide; FEV1, forced expiratory volume in 1 s; SUVmax, maximum standardized uptake value; VATS, video assisted thoracoscopic surgery. ^a^ Includes eye, intestine, omentum, pancreas, scalp, spleen, and perianal soft tissue. ^b^ Includes chemotherapy, immunotherapy, and targeted therapy. ^c^ Pathologic stage determined at the time of primary tumor resection; staging reflects prior treatment to metastatic sites.

**Table 2 cancers-13-01893-t002:** Factors associated with event-free and overall survival (*n =* 111).

Outcome, Variable	Univariable		Multivariable	
	HR (95% CI)	*p*	HR (95% CI)	*p*
Event-free survival				
Pathologic primary tumor size, cm	1.15 (1.03–1.28)	0.013	1.15 (1.03–1.29)	0.014
Lymphovascular invasion	1.85 (1.16–2.96)	0.010	1.73 (1.06–2.84)	0.029
Overall survival				
Receipt of neoadjuvant therapy	0.52 (0.30–0.89)	0.018	0.43 (0.24–0.77)	0.004
Pathologic primary tumor size, cm	1.21 (1.06–1.38)	0.004	1.18 (1.02–1.35)	0.023
Pathologic N1 or N2 disease (vs. N0)	2.05 (1.18–3.56)	0.010	1.83 (1.05–3.20)	0.033
Visceral pleural invasion	2.45 (1.42–4.21)	0.001	1.93 (1.10–3.40)	0.022

CI, confidence interval; HR, hazard ratio. The full univariable analysis for event-free survival (EFS) and overall survival (OS) is included in Appendix A, respectively.

## Data Availability

Data are available from the corresponding author on request.

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
