# Peer review of "Management of Synchronous Extrathoracic Oligometastatic Non-Small Cell Lung Cancer"

_cancers, 2021, doi:10.3390/cancers13081893_

Round 1

Reviewer 1 Report

Gregory D.  et al have described a large number of patients undergoing surgery for synchronous extrathoracic oligometastatic NSCLC. More researches are needed to establish the role of surgery in this type of patients. This is a very interesting paper in the field and well written as well.

Reviewer 2 Report

This study was about management of oligometastatic NSCLC and found surgical resection of the primary tumor was important for long-term survival. It was well-written paper, but I have several minor concerns.

  1. This study showed that there was no difference of survival according to metastasis site. Because of the large proportion of brain metastasis cases in the study, it could affect the result. If the proportion of metastasis sites were well balanced, would the result be different?
  2. In line 142-145, 'Of the patients who underwent LCT for metastases, 76 of 93 (82%) had undergone treatment of metastatic lesions before resection of the primary tumor, ...'.  What do you think of the advantage of this strategy - control of metastasis before primary tumor resection - over opposing sequence?
  3. You concluded in line 339-341 as  '... and those with limited intrathoracic disease are most likely to benefit from a surgery-focused approach ...'. Please present about 'limited intrathoracic disease' in detail. I could not find the results to support this.
  4. In supplementary table 3, what does 'confirmed' mean?

Reviewer 3 Report

This is an interesting analysis about the role of surgery in NSCLC cancer patient with oligometastasis in relatively a small cohort.

Major comments.

  • In method, the authors only included patients with R0 resection as shown in consrt diagrm in Supple Fig.1. However there are analysis data comparing R0 vs R1/R2 in Supple Table 1 and 2. Please explain it.
  • It is surprising that the neoadjuvant chemotherapy itself influenced on OS. However it did not in EFS. Logically the most important factor for OS could be the systemic treatment after progression. And this was not analyzed in this study. And the multivariate analysis in this study could not consider such factors. Please re-compare whether there were differences between two group (Neoadj group 88  patients vs no neo adj group 23 patients).
  • Usually lymphovascular invasion is an important factor for OS. Please show us its HR for OS in multivariate analysis.
  • Are there patients with targetable mutation and got appropriate targeted treatment?
  • Because the participants are patients with oligometastasis, it is inapporpriate to use the term of pathologic stage 0, I, II/III in the table 1.
